# A Coupled Hydrologic–Hydraulic Model (XAJ–HiPIMS) for Flood Simulation

**Yueling Wang** [1]  **and Xiaoliu Yang** [2,*]

[1] Key Laboratory of Water Cycle and Related Land Surface Processes, Institute of Geographic Sciences and Natural Resources Research, Chinese Academy of Sciences, Beijing 100101, China; wangyl@igsnrr.ac.cn

[2] College of Urban and Environmental Sciences, Peking University, Beijing 100871, China

\* Correspondence: xlyang@urban.pku.edu.cn; Tel.: +86-10-64881192

**Abstract:** To protect ecologies and the environment by preventing floods, analysis of the impact of climate change on water requires a tool capable of considering the rainfall-runoff processes on a small scale, for example, 10 m. As has been shown previously, hydrologic models are good at simulating rainfall-runoff processes on a large scale, e.g., over several hundred km$^2$, while hydraulic models are more advantageous for applications on smaller scales. In order to take advantages of these two types of models, this paper coupled a hydrologic model, the Xinanjing model (XAJ), with a hydraulic model, the Graphics Processing Unit (GPU)-accelerated high-performance integrated hydraulic modelling system (HiPIMS). The study was completed in the Misai basin (797 km$^2$), located in Zhejiang Province, China. The coupled XAJ–HiPIMS model was validated against observed flood events. The simulated results agree well with the data observed at the basin outlet. The study proves that a coupled hydrologic and hydraulic model is capable of providing flood information on a small scale for a large basin and shows the potential of the research.

**Keywords:** Xinanjiang model; high-performance integrated hydraulic modelling system (HiPIMS); coupled hydrologic–hydraulic model; two-dimensional surface confluence calculation over basin

---

## 1. Introduction

Flood, as the most common natural hazard, causes huge causalities, heavy economic losses, and ecological and environmental risks worldwide [1,2]. A warmer climate creates more extreme storm events, combined with rapid land use/cover change and a growing population, the risk of floods to urban and farming areas is increasing and posing a threat to ecologies and the environment [2–6]. Extreme rainfall is considered as the main cause of flooding [7], however, inadequate flood risk management also contributes to the increase of flood risk [8]. There is an urgent requirement for flood simulation/forecasting and integrated flood risk management in our changing environment.

Hydrologic models are commonly implemented for rainfall-runoff simulations to predict the basin-scale flow of water passing through the basin outlet, normally on a daily/hourly temporal scale, e.g., Soil and Water Assessment Tool (SWAT) [9,10], Xinanjing (XAJ) [11,12], Stanford Watershed Model (SWM) [13], Sacramento Model (SAC) [14]. However, high spatial heterogeneity of a realistic basin raises the requirement for fine flood management characterized by spatial information inside of the complex basin that affects its response to extreme storms. Numerous distributed grid-based hydrologic models have been developed to represent spatial variability [15–17]. River channel flow is computed by the Muskingum method, e.g., [15], or the Muskingum method is coupled with a one-dimensional (1D) hydraulic model to generate water depth along a river channel, e.g., [18]. The overland flow is barely considered as a fully dynamic flow, and is normally solved for by hydrologic concepts such as linear reservoir [19] or is assumed to be a two-dimensional (2D) kinematic/diffusive wave equation [16].

Hence, the flood characteristics (i.e., flood extent, water depth, and velocity) cannot be represented inside the two-dimensional domain.

Currently, the two-dimensional hydraulic model provides a good solution for flood routing processes for fine flood risk assessment and management [20–22]. The fully 2D hydraulic model is a robust numerical methodology to provide accurate simulation of runoff flow on a minutely temporal scale, based on high-resolution data sources and GPU-accelerated techniques [23–25]. However, runoff generation is still a big concern for the application of the fully 2D hydraulic model in the large-scale of a basin, especially for small and medium floods. The typical solution of infiltration is to adopt the Kostiakov equation (e.g., [26]), Green–Ampt model (e.g., [27]), Horton equation (e.g., [28,29]), or Richard model (e.g., [30]).

In recent years, how to jointly use the hydrologic and hydraulic models for flood risk assessment and reservoir operation has been discussed [31–33]. As has been shown previously, hydrologic models are good at simulating rainfall-runoff processes for a basin on a large scale, e.g., several hundred km$^2$, while hydraulic models are more advantageous for applications on smaller scales. In order to take advantage of these two types of models, this study attempts to couple a hydrologic model with a hydraulic model.

In this study, the Misai river basin (797 km$^2$), located in Zhejiang Province, China, is taken as the study area. For the hydrologic model, XAJ was selected as it has been widely used in China for a long time [34–36]. For the hydraulic model, the high-performance integrated hydraulic modelling system (HiPIMS) was chosen because of its high-speed calculations and wide application [23–25]. This paper introduces how the XAJ–HiPIMS is coupled from XAJ and HiPIMS and validates the new model against typical floods. Section 2 outlines the materials and methods, Section 3 presents results and discussion, and Section 4 gives conclusions.

## 2. Materials and Methods

### 2.1. XAJ

XAJ is a lumped rainfall-runoff model developed by the authors of [11,12]. Based on the watershed saturation-excess runoff theory, evapotranspiration is calculated in three soil layers. The soil's moisture storage capacity distribution curve is utilized to provide a non-uniform distribution of storage capacity in the hydrologic unit. The total runoff is composed of surface runoff, interflow, and groundwater runoff. The distribution of these three components can be decided by using a free water capacity distribution curve. The surface runoff is routed by the unit hydrograph. The interflow and groundwater flow are routed by the linear reservoir method. Hence, four modules compose the XAJ: evapotranspiration, runoff generation, runoff sources partition, and runoff routing ([11,12], Figure 1). The parameters of XAJ are described in Table 3. The output of XAJ is outflow discharge at the outlet section of the basin.

### 2.2. HiPIMS

The governing equations of HiPIMS are fully 2D shallow water equations based on the Cartesian uniform grid [38], as follows,

$$\frac{\partial q}{\partial t} + \frac{\partial f}{\partial x} + \frac{\partial g}{\partial y} = R + S_b + S_f \tag{1}$$

$$q = \begin{bmatrix} h \\ uh \\ vh \end{bmatrix}, f = \begin{bmatrix} uh \\ u^2h + \frac{1}{2}gh^2 \\ uvh \end{bmatrix}, g = \begin{bmatrix} vh \\ uvh \\ v^2h + \frac{1}{2}gh^2 \end{bmatrix}$$

$$R = \begin{bmatrix} r \\ 0 \\ 0 \end{bmatrix}, S_b = \begin{bmatrix} 0 \\ -gh\partial b/\partial x \\ -gh\partial b/\partial y \end{bmatrix}, S_f = \begin{bmatrix} 0 \\ -\tau_{bx}/\rho \\ -\tau_{by}/\rho \end{bmatrix} \tag{2}$$

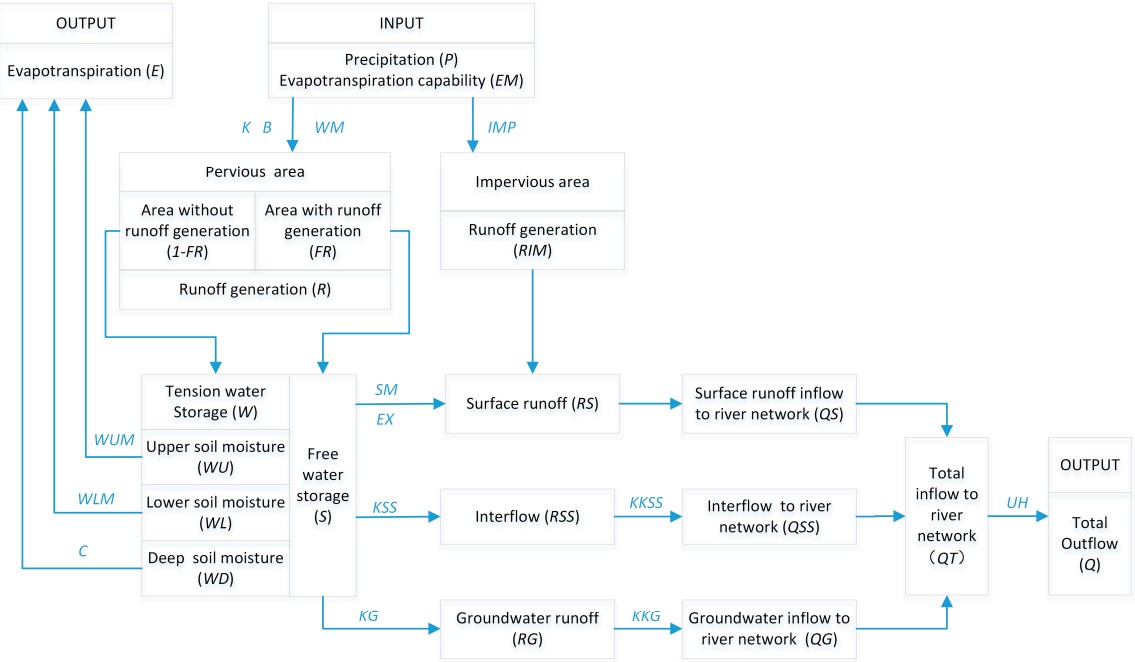

**Figure 1.** The framework of the Xinanjing model (XAJ) [37].

In Equation (1), $q$ stands for the vector of conserved flow variables; $f$ and $g$ are the flux vectors in the x- and y-directions, respectively; and R, $S_b$, and $S_f$ denote the source terms of rainfall, bed slope, and friction, respectively. In Equation (2), $h$ is the water depth ($h = \eta - z_b$, where $\eta$ is the water surface elevation and $z_b$ is the bed elevation); $u$ and $v$ are the depth-averaged velocity components in the x- and y-directions, respectively; $g$ is the acceleration of gravity; $\rho$ is the water density; $r$ is the generated surface runoff (which is the $r_{i,j}^n$ in Equation (2)); $\partial b / \partial x$ and $\partial b / \partial y$ denote the bed slope in the x- and y-directions; $\tau_{bx}$ and $\tau_{by}$ are the bed friction stresses, calculated by the following equations:

$$\tau_{bx} = \rho C_f u \sqrt{u^2 + v^2} \ \ and \ \ \tau_{by} = \rho C_f v \sqrt{u^2 + v^2} \tag{3}$$

where $C_f \ ( = g n^2 / h^{1/3} )$ stands for the bed roughness coefficient; and $n$ is the Manning's coefficient.

In this study, HiPIMS employs a first-order finite volume Godunov-type numerical scheme to solve the above governing equations. Equation (1) is discretized by Equation (4) and the flow variables can be updated as follows.

$$q_i^{m+1} = q_i^m - \frac{\Delta t}{\Omega_i} F_k(q^m) l_k + \Delta t \left( R_i^m + S_{bi}^m + S_{fi}^{m+1} \right),$$
$$F_k(q) = f_k(q) n_x + g_k(q) n_y, \tag{4}$$

where the superscript $m$ denotes the present time level; $\Delta t$ is the present time step (which is the $\Delta t^n$ in Equation (5)); $\Omega_i$ stands for the area of cell $i$; $F_k(q)$ presents the fluxes that are normal to the cell edges; $k$ is the index of the cell edges ($k$ = 1–4); $l_k$ is the corresponding cell edge length; and $n = (n_x, n_y)$ denotes the unit vector of the outward normal direction. In HiPIMS, the flux terms and bed slope term are calculated by an explicit scheme. The interface fluxes are calculated by the HLLC (Harten–Lax–van Leer-Contact) approximate Riemann solver. The local Riemann problems at the cell interfaces are solved by implementing the surface reconstruction method [39]. The friction term is solved by an implicit scheme [40], which is capable of maintaining the numerical stability when dealing with very small water depth. The adaptive time step method is implemented and controlled by the Courant–Friedrichs–Lewy (CFL) criterion. The high-performance of this CPU/GPU-integrating model is achieved by adopting the OpenCL programming framework [38,41].

## 2.3. Coupling Framework

In the coupling framework, XAJ generates the surface runoff and produces interflow and underground flow while the hydraulic model is run for 2D confluence calculations of the surface runoff generated by XAJ. The generated surface runoff was presented in the governing equations of HiPIMS as the source term in the mass conservation equation. The temporal scale of XAJ is hourly while the computing time-step of HiPIMS is in seconds, and the spatial scale of XAJ is the whole basin, while computing grid-cell of HiPIMS is in meters. In order to deal with this temporal and spatial downscaling issue, functions (Equations (5) and (6)) were created as below.

$$r_{i,j}^* = R_{XAJ} \times \frac{\Delta t^n}{3600} \tag{5}$$

$$r_{i,j}^n = r_{i,j}^* / m \tag{6}$$

In Equation (5), $R_{XAJ}$ denotes the surface runoff calculated by the XAJ; $\Delta t^n$ is the time-step of HiPIMS; $n$ means the presently considered time-step; and $r_{i,j}^*$ stands for the downscaled surface runoff. In Equation (6), $r_{i,j}^n$ means the surface runoff in terms of the source term in the mass conservation equation of HiPIMS and $m$ is the amount of the computing grid cell.

Then, the surface runoff generated by XAJ inputs to the hydraulic model to produce two-dimensional flow over the basin. Thus, the procedure is proposed as follows (shown in Figure 2): (1) Calculate the surface runoff by XAJ with the temporal scale of 1 h; (2) Allocate the surface runoff using Equation (5) in order to transform the temporal scale from hours to seconds; (3) Distribute the surface runoff into each computing grid cell of HiPIMS using Equation (6); (4) Compute the dynamic surface flow process and produce the spatial distribution of the inundated extent, water depth, and velocity in the whole basin by HiPIMS; (5) Overlay the surface flow process at the basin outlet produced by HiPIMS and the processes of interflow and underground flow given by XAJ to form the simulated flow process at the basin outlet and validate it against the flow observed at the outlet.

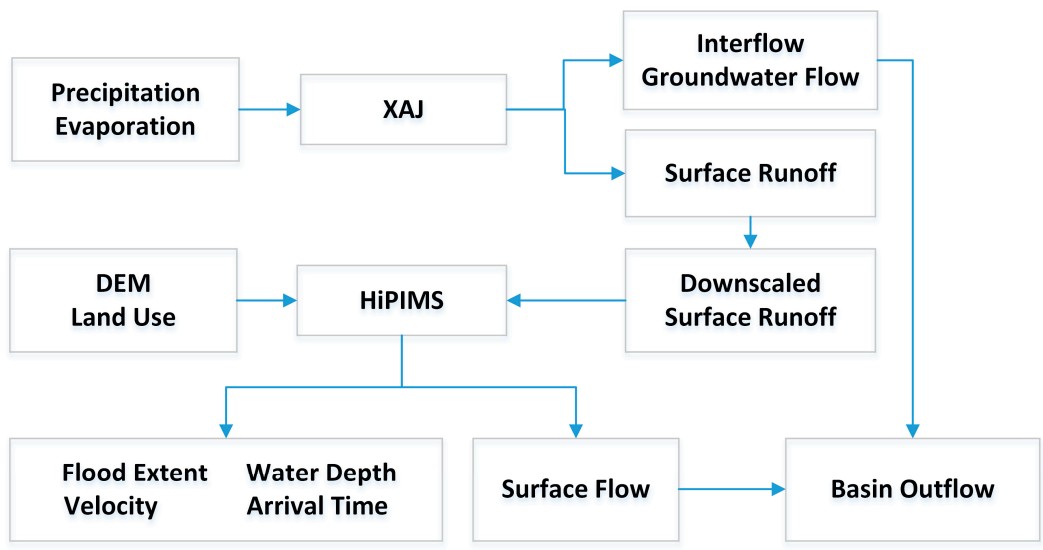

**Figure 2.** Flowchart of the coupling framework.

## 2.4. Statistical Method

In this work, the statistical methods absolute relative error of flood-peak discharge (ARED) (Equation (7)), difference of peak arrival time (DPAT) (Equation (8)), and the Nash–Sutcliffe efficiency coefficient (NSE) (Equations (9) and (10)) are employed to estimate the discrepancy of the discharge. Three types of data are considered herein, i.e., observation, simulation by XAJ, and simulation by XAJ–HiPIMS.

$$\text{ARED} = \frac{\left| \text{MAX}(Q_s) - \text{MAX}(Q_o) \right|}{\text{MAX}(Q_o)} \tag{7}$$

where $\text{MAX}(Q_s)$ stands for the simulated peak discharge and $\text{MAX}(Q_o)$ denotes the observed peak discharge. $\text{ARED} \in (-\infty, +\infty)$,

$$\text{DPAT} = T_{\text{MAX}(Qs)} - T_{\text{MAX}(Qo)} \tag{8}$$

where $T_{\text{MAX}(Qs)}$ is the arrival time of simulated peak discharge by XAJ and XAJ–HiPIMS and $T_{\text{MAX}(Qo)}$ is the arrival time of the observed peak discharge.

$$NSE_1 = 1 - \frac{\Sigma (Q_o - Q_1)^2}{\Sigma (Q_o - \overline{Q_o})^2} \tag{9}$$

$$NSE_2 = 1 - \frac{\Sigma (Q_o - Q_2)^2}{\Sigma (Q_o - \overline{Q_o})^2} \tag{10}$$

where $NSE_1$ is the Nash–Sutcliffe efficiency coefficient computed by the XAJ simulation and observation; $NSE_2$ is the Nash–Sutcliffe efficiency coefficient computed by the XAJ–HiPIMS simulation and observation; $Q_o$ is the observed discharge; $Q_1$ is the simulated discharge of XAJ; $Q_2$ is the simulated discharge of XAJ-HiPIMS; and $\overline{Q_o}$ is the averaged observed discharge.

### 2.5. Study Area

The Misai basin is located in the Zhejiang province of China with an area of 797 km$^2$. The bed elevation differs from 105 to 1253 m. In this study, the 30-m ASTER GDEM V2 elevation data were adopted for computing, as shown in Figure 3. Six precipitation gauging stations are located in this basin. One is located at the outlet of the basin, and it also served as stream gauging station. The 30-m Landsat TM image data were used to provide the land use information. Considering the land characteristics, the land use types were classified into seven categories in this work, i.e., forest, heavy brush, cultivated land, grass land, pond/river, bare land, and urban land (distribution shown in Figure 4). The proportional area and Manning's coefficients of land use types are summarized in Table 1.

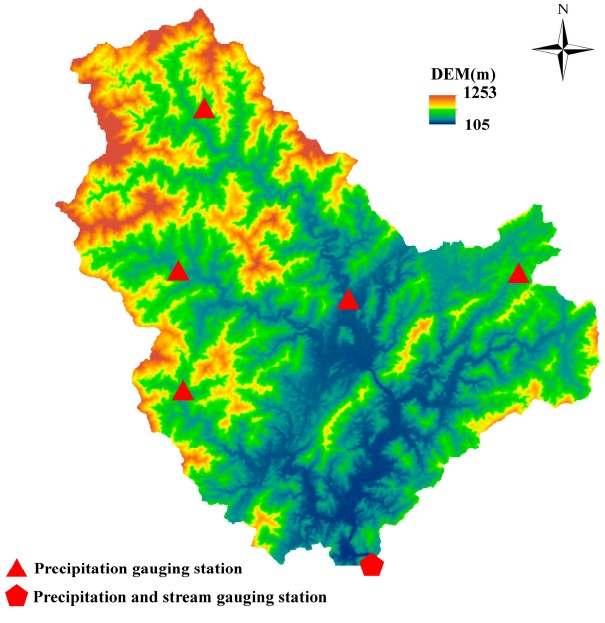

**Figure 3.** Map of the Misai basin.

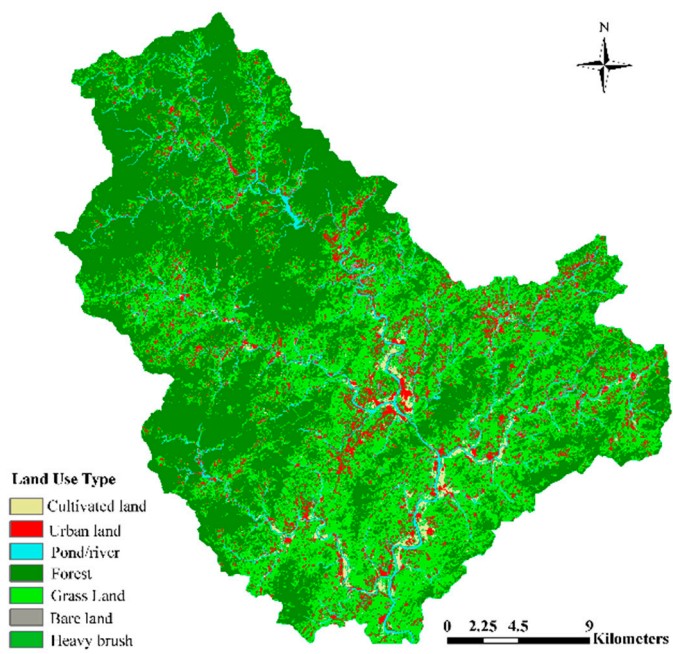

**Figure 4.** Land use of the Misai basin.

**Table 1.** The proportional area and Manning's *n* of different land use types in the Misai basin.

| No. | Land Use Type | Area Proportion (%) | Manning's *n* [1] |
|-----|---------------|---------------------|-------------------|
| L1 | Forest | 45.5 | 0.139 |
| L2 | Heavy brush | 37.3 | 0.098 |
| L3 | Cultivated land | 2.2 | 0.041 |
| L4 | Grass land | 7.3 | 0.031 |
| L5 | Pond/river | 5.8 | 0.021 |
| L6 | Bare land | 0.7 | 0.026 |
| L7 | Urban land | 1.2 | 0.013 |

[1] summarized from [42–44].

### 2.6. Flood Processes

Three typical flood processes (FPs) were chosen to represent big (B: peak discharge > 1000 m$^3$/s), medium (M: 500 m$^3$/s < peak discharge < 1000 m$^3$/s), and small (S: peak discharge < 500 m$^3$/s) floods, as shown in Table 2.

**Table 2.** The three typical flood processes (FPs).

| Name | Start Time | End Time | Rainfall Height | Peak Flow |
|------|-----------|----------|-----------------|-----------|
| FP(S) | 1987/5/26 8:00 | 1987/5/27 20:00 | 65 mm | 202 m$^3$/s |
| FP(M) | 1985/5/5 8:00 | 1985/5/7 12:00 | 106 mm | 708 m$^3$/s |
| FP(B) | 1983/5/29 8:00 | 1983/5/30 13:00 | 222 mm | 1820 m$^3$/s |

### 2.7. Modelling Set

XAJ: The rainfall input was given at 1-h intervals from the historic records from the six gauging stations in the basin. The 13 coefficients were calibrated and are shown in Table 3.

HiPIMS: The considered basin was divided into 909,824 grid cells upon a regular uniform grid of resolution 30 m. The Manning's *n* are given in Table 1. The CFL number was set to 0.35 in the present work.

**Table 3.** The parameters of XAJ.

| Module | Parameters | Physical Meaning | Value |
|---|---|---|---|
| Evapotranspiration | WUM | Averaged soil moisture storage capacity of the upper layer | 14 |
| | WLM | Averaged soil moisture storage capacity of the lower layer | 86 |
| | WDM | Averaged soil moisture storage capacity of the deep layer | 33 |
| | K | Conversion coefficient of evaporation | 1 |
| | C | Coefficient of the deep layer | 0.126 |
| Runoff generation | B | Exponential of the distribution to tension water capacity | 0.375 |
| | IMP | Percentage of impervious and saturated areas in the catchment | 10 |
| Runoff source partition | SM | Areal mean free water capacity of the surface soil layer | 97 |
| | EX | Exponent of the free water capacity curve influencing the development of the saturated area | 1.03 |
| | KG | Outflow coefficients of the free water storage to groundwater relationships | 0.459 |
| | KSS | Outflow coefficients of the free water storage to interflow relationships | 0.07 |
| Runoff routing | KKG | Recession constants of the groundwater storage | 0.997 |
| | KKSS | Recession constants of the lower interflow storage | 0.747 |

## 3. Results and Discussion

In this study, the simulations of XAJ–HiPIMS and XAJ were assessed by comparing their results with the observed discharge at the outlet gauging station, as presented in Figure 5. Good agreement of the simulation with the observation was demonstrated for the three typical floods. Better performance was found in the bigger flood category.

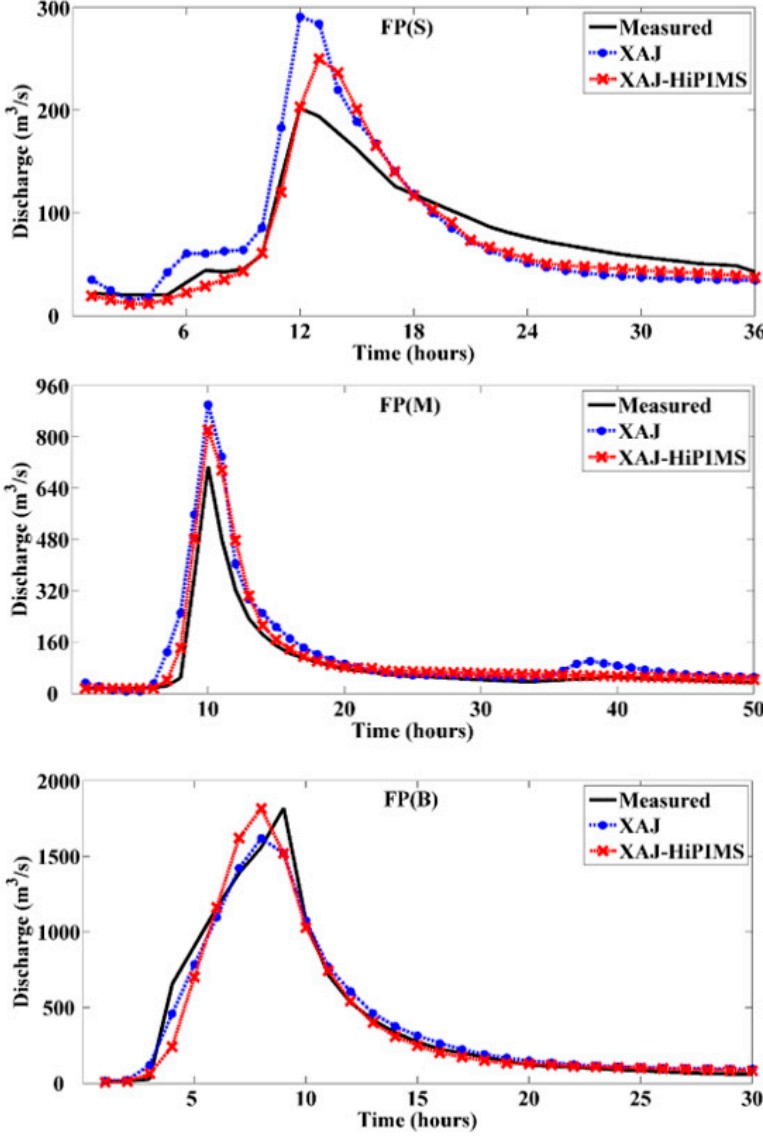

**Figure 5.** Time history of discharge at the outlet gauging station.

Further assessment has been done by implementing the statistical methods of ARED, DPAT, and NSE, shown in Table 4. We found that (1) in terms of the peak discharge, the simulation of XAJ–HiPIMS presented good agreement with the measurement; (2) in terms of ARED, the XAJ–HiPIMS achieved a better flood peak simulation, especially for the bigger flood; (3) in terms of DPAT, the differences between simulations and measurements were no more than 1 h and smaller differences were found in the medium flood; (4) in terms of NSE, XAJ–HiPIMS obtained good flood simulations, which were greater than 0.85. The above results demonstrate XAJ–HiPIMS could provide satisfactory simulation at different flood scales.

**Table 4.** The statistical analysis of the measured and simulated discharge (Q).

| FP No. | Peak discharge (m$^3$/s) | | | ARED (%) | | DPAT (hour) | | NSE | |
|---|---|---|---|---|---|---|---|---|---|
| | O | H | C | H | C | H | C | $NSE_1$ | $NSE_2$ |
| FP(S) | 202 | 291 | 250 | 0.44 | 0.24 | 0 | 1 | 0.6392 | 0.8459 |
| FP(M) | 708 | 900 | 822 | 0.27 | 0.16 | 0 | 0 | 0.7111 | 0.8524 |
| FP(B) | 1820 | 1620 | 1817 | 0.11 | 0.002 | −1 | −1 | 0.9757 | 0.9422 |

O: Observed data; H: Simulated by XAJ; C: Simulated by XAJ–HiPIMS.

According to the 2D distribution of flood information, inundated extent and maximum water depth were investigated. The threshold of inundated depth was set to 0.01 m in this study. The snapshots of inundated extent were produced by XAJ–HiPIMS at the peak moment (the moment of peak discharge) shown in Figure 6 (left column). The value of water depth in each grid cell was recorded during the calculation. Maximum water depth is defined as the maximum value of water depth in each grid cell in the entire flood event record. The maps of maximum water depth were analyzed and are presented in Figure 6 (right column).

We found that: (1) Inundated area was mostly located along the river network and in the low-lying urban areas and cultivated land. Inundated extent and maximum water depth were obviously increased due to the bigger rainfall height. (2) In three FPs, maximum water depth in low-lying areas of urban land exceed 1 meter. In FP(B), cultivated land was also inundated with the maximum water depth of more than 1 meter. The results suggest that spatial distribution of flood information could be adopted in flood risk assessment and also help with the planning of flooding mitigation and adaptation strategies.

These findings imply that the XAJ–HiPIMS (1) presents good agreement with the observations; (2) can provide 2D presentation of flood processes and more information of flood characteristics at the basin scale; (3) provides a new perspective and methodology for ecological and environmental protection, flood prevention and management, and urban water landscape design.

## 4. Conclusions

This study presented a coupled hydrologic–hydraulic model, XAJ–HiPIMS. The model was validated in the Misai basin, Zhejiang Province, China by comparing its performance with measurements and statistically assessing the absolute relative error of flood-peak discharge, the difference of peak arrival time, and the Nash–Sutcliffe efficiency coefficient.

The present study: (1) demonstrates that the XAJ–HiPIMS can provide a good performance in basin-scale flood simulation and deliver a 2D viewpoint of flood characteristics; (2) reveals the potential application of XAJ–HiPIMS in ecological and environmental protection, urban flood management, landscape design, and climate change analysis; and (3) brings forward the necessity and recommendations for further research.

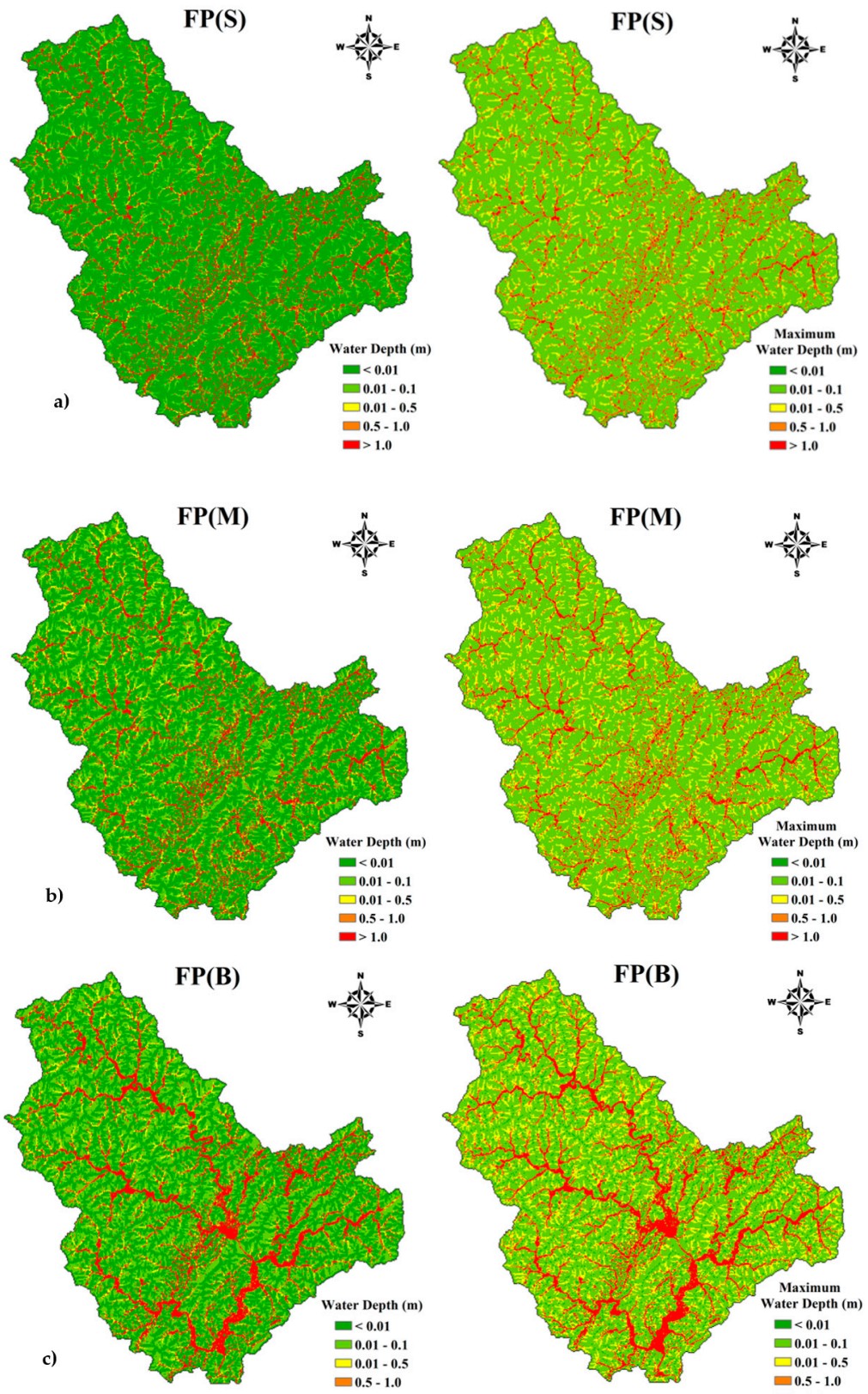

**Figure 6.** (**a**) FP(S): Snapshot of inundated extent at the peak moment (left) and map of maximum water depth (right); (**b**) FP(M): Snapshot of inundated extent at the peak moment (left) and map of maximum water depth (right); (**c**) FP(B): Snapshot of inundated extent at the peak moment (left) and map of maximum water depth (right).

**Author Contributions:** All authors have read and agreed to the published version of the manuscript. Conceptualization, Y.W. and X.Y.; methodology, Y.W. and X.Y.; software, Y.W. and X.Y.; validation, Y.W.; formal analysis, Y.W.; investigation, Y.W. and X.Y.; resources, Y.W. and X.Y.; data curation, Y.W.; writing—original draft preparation, Y.W.; writing—review and editing, Y.W. and X.Y.; visualization, Y.W.; supervision, X.Y.; project administration, X.Y.; funding acquisition, Y.W.

**Funding:** This research was funded by the National Natural Science Foundation of China, grant numbers 41890823 and 51609231.

**Acknowledgments:** The authors would like to acknowledge Liang Qiuhua of Loughborough University of UK for kindly providing the HiPIMS software used in this research, and Huili Chen of Loughborough University of UK for her interpretation of land use information, and Chaowei Xu of Peking University for his helping in XAJ use. Special thanks go to the three anonymous reviewers for their comments and suggestions, valuable and helpful in improving the quality of this paper.

**Conflicts of Interest:** The authors declare no conflict of interest. The funders had no role in the design of the study; in the collection, analyses, or interpretation of data; in the writing of the manuscript, or in the decision to publish the results.

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
