# Peer review of "A Coupled Hydrologic–Hydraulic Model (XAJ–HiPIMS) for Flood Simulation"

_water, doi:10.3390/w12051288_

Round 1
Reviewer 1 Report
This is a potentially interesting contribution regarding flood modelling in cathments. I thing that the paper would require significant improvement before publication.
The presention is sometimes unclear. In particuler the presentation of the hydrologic model is not good :
- what is the stored-full runoff theory ?
- how can the output of a distributed hydrological model be one-dimensional (which dimension are you talking about ?) ? In equation (5), it appears the computed runoff from XAJ writes rxi,j , which suggests a bi-dimensional output for the model.
- Providing the chosen (or partly optimized, it is certainly not possible to optimize 13 parameters) parameters of XAJ (table 3) is useless when the reader knows nothing regarding the base equations of the model.
- How was XAJ calibrated ?
The description of coupling must also be improved :
- More should be said about the coupling of the models. Is there some routing procedure for runoff in XAJ ? If yes, how can be combined runoff routing and 2D St Venant simulations ? If non, how can the XAJ model provide flow simulation at the outlet ?
- How are routed interflow and groundwater runoff ? (which dynamics, which directions, distributed routing or not ?)
- How is miwed runoff by XAJ and flow simulation by HiPIMS ?
The coupled model provides interesting distributed information. But there is no validation of this information. This is needed. For exemple, for the largest flood, it appears (table 6) that about 300 km2, i.e. one third of the catchment area, was inundated. There must be a way to validate how realistic this (so strong) conclusion is. More generaly distributed data regarding water levels or inundation maps (even if not very precise) would be most useful for the validation of the coupled model.
The obtained results (figure 4) show that the introduction of the hydraulic model HiPIMS improves the simulation. To evaluate this improvment, it would be necessary to make sure that the obtained simulation with the hydrologic model XAJ was the best possible simulation. This should be argued.
Miscellaneous :
- figure 6 (and 7, bad numbering) are not readable.
- in figure 2, please revert the color scale and user brown for the highest altitude and blue for the lowest.
- I would not call equation (5) downscaling, this is only a flat distribution
Reviewer 2 Report
The manuscript reports on the use of a Hydrologic-Hydraulic coupled model for realistic flood simulations. The models were previously developed and validated.
1. The abstract and introduction also mention the use of GPU technology but nothing is indicated about computational times when presenting the results.
2. The literature review on this type of coupled models is short. It should be improved.
3. How is the computational grid? Is it a rectangular or a quadrilateral grid?
4. Fig. 6 is not clear/useful and is repeated twice.
5. How is the computational time increased by the hydraulic model over that of the hydrologic model? The results are only slightly improved.
Reviewer 3 Report
Dear authors,
you present a nice and pragmatic solution to couple a "simple" hydrologic model with 2D flood modelling.
However the representation of your work requires improvement. Currently, the hydrologic model and its application for the case study is not clear. Namely, the spatial resolution of the "lumped" model needs be explained and shown on the map. The method of runoff generation with the coupled model is for me not completely clear. The discusion of the results should be performed with a more critical view to the own work. The method has several clear limitations due to the simple downscaling and the uniform coupling (e.g. infiltration due to overland flow will not be feasible). You should be aware of these and clearly derive suggestion, under which conditions your method is applicable and where not.
Please consider all remarks I made in the pdf.
English is ok, so far, but I noticed serveral grammar errors, namely singular/plural do not always conform with the conjugated form of the verb.

Round 2
Reviewer 2 Report
The authors have improved the manuscript according to my questions. I am ready to recommend it for publication.
Reviewer 3 Report
The authors answered thouroughly to the comments and questions. They improved the paper by adding missing information, supported by graphical schemes of the model. Now I understood that it is really a lumped model of the whole area, lineary downscaled to a 30x30 m grid. Actually, this is the combination of two extremes. In future, it requires more research to assess and eventually reduce the uncertainty of this approach.
However, the idea is really nice and could provide a quick and good estimate of flooding in rather rural areas.